# Textural or Textual: How Vision-Language Models Read Text in Images

**Hanzhang Wang**[1][*]   **Qingyuan Ma**[1]

## Abstract

Typographic attacks are often attributed to the ability of multimodal pre-trained models to fuse textual semantics into visual representations, yet the mechanisms and locus of such interference remain unclear. We examine whether such models genuinely encode textual semantics or primarily rely on texture-based visual features. To disentangle orthographic form from meaning, we introduce the ToT dataset, which includes controlled word pairs that either share semantics with distinct appearances (synonyms) or share appearance with differing semantics (paronyms). A layer-wise analysis of Intrinsic Dimension (ID) reveals that early layers exhibit competing dynamics between orthographic and semantic representations. In later layers, semantic accuracy increases as ID decreases, but this improvement largely stems from orthographic disambiguation. Notably, clear semantic differentiation emerges only in the final block, challenging the common assumption that semantic understanding is progressively constructed across depth. These findings reveal how current vision-language models construct text representations through texture-dependent processes, prompting a reconsideration of the gap between visual perception and semantic understanding. The code is available at: https://github.com/Ovsia/Textural-or-Textual

## 1. Introduction

While vision-language models have demonstrated the capacity to process textual content embedded within images, a fundamental question remains: *do they genuinely capture the semantic meaning of the text, or merely treat it as another visual pattern?* This distinction is especially pertinent

given that textual elements, while structurally and symbolically distinct from visual objects, may nevertheless be encoded in ways that resemble general visual pattern recognition rather than language-specific processing. This echoes findings in cognitive neuroscience suggesting that the brain regions supporting visual word recognition (e.g., the visual word form area) are recycled from object recognition systems and may initially process written words similarly to visual objects before being modulated by linguistic feedback (Carreiras et al., 2014). Such a tendency raises concerns that these models may be aligning low-level textures rather than achieving a deeper, cross-modal semantic integration. Moreover, given the hierarchical nature of neural network processing, it remains unclear at what stage textual features begin to influence an image's semantic interpretation. These challenges motivate our study, which aims to disentangle textual and textural representations to better understand how vision-language models encode and form textual semantics.

One significant manifestation of these uncertainties is typographic attacks (Goh et al., 2021), which highlight vulnerabilities in vision-language models when interpreting text within images. These attacks embed misleading text into images, causing misclassifications driven by the misalignment between textual and visual modalities. For instance, an image of a dog overlaid with the word "laptop" may be misclassified as an electronic device (Lemesle et al., 2022). As models like GPT-4v (Yang et al., 2023) become increasingly capable, these vulnerabilities raise pressing security concerns, including the risk of unintended command execution resembling model "jailbreaking" (Gong et al., 2023; Robey et al., 2023; Wang et al., 2023). Addressing such vulnerabilities requires a deeper understanding of how typographic attacks exploit the models' internal representations.

Although typographic attacks may not conform to traditional definitions of adversarial perturbations, they nonetheless expose how vision-language models can conflate visual text with image content, reflecting a fragile alignment between symbolic and perceptual representations. These models often exhibit implicit associations between textual cues and corresponding visual concepts (Cao et al., 2023), suggesting a shared embedding space where semantically related modalities are co-located. For instance, a model might position the visual features of a cat near both the textual token "cat" and its associated conceptual representation. However, it

[1]School of Computer Engineering and Science, Shanghai University, Shanghai, China. Correspondence to: Hanzhang Wang <hanzhang.mon.wang@gmail.com>.

*Proceedings of the $42^{nd}$ International Conference on Machine Learning*, Vancouver, Canada. PMLR 267, 2025. Copyright 2025 by the author(s).

remains unclear whether such proximity reflects genuine semantic understanding or arises from superficial correlations learned during pretraining. Disentangling these factors is essential for understanding the robustness and interpretability of vision-language representations under multimodal perturbations.

These observations raise an important question: to what extent do vision-language models distinguish between the visual appearance of text and its semantic content? To explore this, we use Intrinsic Dimension (ID) to quantify the complexity of visual representations by measuring the degrees of freedom required to encode them accurately. Rooted in high-dimensional geometry, ID has been widely applied to the analysis of neural network behavior, particularly in the context of adversarial robustness (Amsaleg et al., 2017; Ma et al., 2018). We extend this approach by applying ID to the ToT dataset, which introduces controlled variations in both orthographic form and semantic meaning. This allows us to examine how visual and conceptual sources of complexity affect the internal representations learned by vision-language models.

We introduce the ToT (Textural or Textual) typographic attack dataset, which contains text overlays that are semantically consistent, irrelevant, or nonsensical with respect to the underlying image. This setup allows us to examine how a pre-trained vision-language model responds to different types of text-image relationships. We further introduce 10 paronym-synonym pairs designed to disentangle how the model represents orthographic similarity versus semantic relatedness. Our analysis shows a non-linear progression in representational dynamics: earlier layers refine texture-level features that facilitate surface-level text recognition, with semantic distinctions emerging only in the final block of the network. Specifically, the main contributions of this work are as follows:

- We provide a systematic analysis of how vision-language models process typographic attacks, using intrinsic dimension (ID) to quantify representational complexity across network layers. Our results show that improvements in semantic decoding accuracy are largely attributable to orthographic recognition, which relies on texture-based features rather than abstract semantic understanding. This challenges the common assumption that semantic representations are gradually constructed across layers. Clear semantic differentiation emerges only at the final block.

- Building on these observations, we defend against typographic attacks by simply fine-tuning only the final block of the model to better distinguish between textural and textual representations. Experimental results show that our strategy effectively balances the performance between the original image and the typographic classification, achieving significant improvements across diverse defense scenarios.

## 2. Related Work

### 2.1. Typographic Attacks

CLIP (Radford et al., 2021) is known for its ability to joint understanding of language and vision. Due to its large amount and spin of training images, many of which incorporate both visual and textual features, it can read visually presented text, or scene-text (Materzyńska et al., 2022; Cao et al., 2023). A notable aspect of CLIP is its tendency, in certain instances, to rely predominantly on text for image classification. This reliance can lead to what's termed a typographic attack (Goh et al., 2021), where misclassification occurs due to overemphasis on text.

In response to such vulnerabilities, various defense strategies have been explored. (Materzyńska et al., 2022) implement a linear transformation to bifurcate the model into two distinct streams: one dedicated to visual information and the other to textual data. (Azuma & Matsui, 2023) introduce the Dense-Prefix token in conjunction with prompt learning, placing it before class names to significantly enhance accuracy against real-world typographic attack scenarios. PAINT (Ilharco et al., 2022) involves a method that interpolates between a model's pre- and post-fine-tuning weights, showing notable success in mitigating typographic attacks. (Cao et al., 2023) takes a different route by filtering out dataset samples containing text regions within images, leading to not only improved defense against typographic attacks but also heightened accuracy in other tasks.

### 2.2. Disentangling Visual and Textual Semantics in Vision-Language Models

Large vision-and-language pre-trained models like CLIP (Radford et al., 2021) showcase their efficacy through extensive pre-training on diverse datasets, excelling in tasks such as image classification (Zhang et al., 2022), visual question answering (VQA) (Song et al., 2022), and image captioning (Mokady et al., 2021). The treatment of visually presented text within these models sparks debate in the field. Some researchers recommend removing the language representation from the visual aspects of the model (Materzyńska et al., 2022; Cao et al., 2023). In contrast, other researchers underscore the indispensable role of language comprehension in tasks like Text-VQA and Text-Captioning (Yang et al., 2021; Kil et al., 2023). They advocate for a harmonious integration of visual and textual information, pointing out that such synergy is crucial for a more holistic understanding of images.

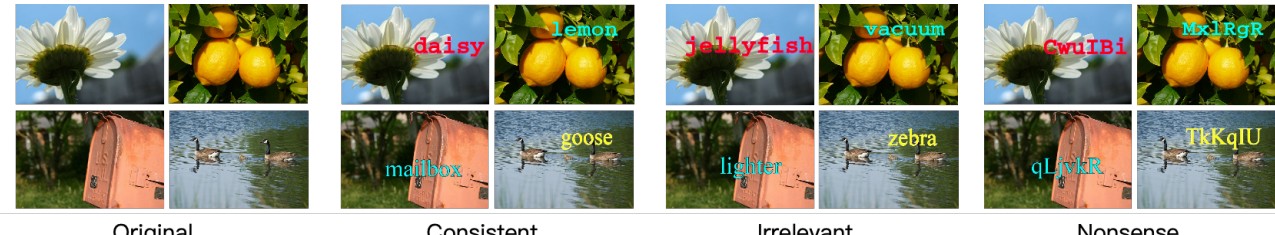

Figure 1. Example images from Subset 1 of the ToT dataset, illustrating consistent, irrelevant, and semantically nonsensical image-text pairs.

In line with this debate, our research undertakes a series of comparative experiments focusing on CLIP's Vision Transformer (Dosovitskiy et al., 2020). These experiments aim to unravel the intricate dynamics between scene-text recognition and the multi-modal properties inherent in CLIP. Addressing the complexities of multi-modal models, particularly their challenge in differentiating visual elements from textual semantics, our study seeks to fine-tune this delicate balance. We endeavor to enhance the model's capability to discern physical objects from scene text, thereby enriching its understanding and interpretation of both visual and textual components in a unified and coherent manner.

## 2.3. Intrinsic Dimensions of Multimodal Representation

The Intrinsic Dimension (ID) is the minimum number of dimensions required to represent data effectively (Levina & Bickel, 2004). In neural networks, ID is derived from the model's representations, indicating the fewest parameters needed to capture specific features (Amsaleg et al., 2015). Ansuini et al. (2019) demonstrated a correlation between the final layer's ID and the model's accuracy, noting that ID typically follows a hunchback-shaped curve across layers, reflecting the learning process (Ansuini et al., 2019). Moreover, ID is crucial for interpreting learned representations and exploring its relationship with neural network training (Aghajanyan et al., 2020; Pope et al., 2021). Basile et al. (2025) introduced a correlation-based metric that uses local intrinsic dimensionality to reveal non-obvious connections between paired visual and textual features, which standard distance or similarity metrics fail to capture.

Amsaleg et al. (2017) and Ma et al. (2018) used local ID to assess adversarial robustness, finding that LID increases with noise in adversarial perturbations. This connection emphasizes how ID influences a model's vulnerability. Tulchinskii et al. (2024) further explored ID in textual data, revealing that human-generated texts have an average ID of 7 to 9, while AI-generated texts often fall below 1.5. This distinction enables classifiers to effectively differentiate between human and AI-generated content.

## 3. Method

### 3.1. ToT Datasets

#### 3.1.1. SUBSET 1: SEMANTIC CONFUSION

We propose the ToT (Textural or Textual) dataset, derived from ImageNet-1k, which features 100 categories of common objects overlaid with texts of varying semantics. The dataset contains 50,000 images, with 500 randomly selected images per category. These categories represent frequently encountered real-world objects with short, distinct names and minimal semantic overlap, making the dataset highly relevant for studying typographic attack scenarios in practical contexts. Figure 1 illustrates the three types of textual modifications applied to the images to generate a diverse set of compositions.

The **Original** subset consists of unaltered images from ImageNet-1k, serving as a baseline. In the **Consistent** subset, each image is overlaid with its matched category label, preserving semantic alignment. The **Irrelevant** subset pairs each image with a randomly selected label from a mismatched ImageNet-1k category, ensuring the overlaid text is a valid label but semantically irrelevant to the image content. The **Nonsense** subset overlays each image with a string formed from random combinations of letters, averaging six characters in length and resembling the structure of category names, but carrying no semantic meaning.

#### 3.1.2. SUBSET 2: PARONYMS VS. SYNONYMS CONFUSION

Since the form of a word is often intrinsically linked to its meaning, variations in word structure typically lead to words with distinct semantic differences. This suggests that neural networks may distinguish words based solely on superficial textural features, leading to what appears to be semantic-level comprehension. To explore this hypothesis, we design a subset of 10-word pairs specifically aimed at disentangling the relationship between word form and meaning.

This subset explores how models differentiate between words that are visually similar but semantically distinct,

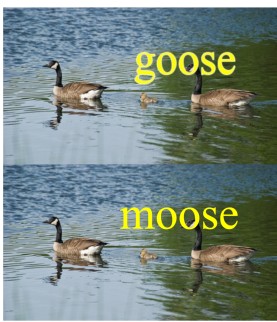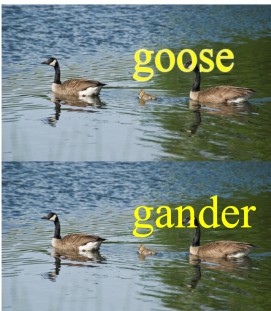

Paronyms Pair          Synonyms Pair

*Figure 2.* Image-text examples from Subset 2 of the ToT dataset with paronym and synonym word pairs, designed to isolate orthographic and semantic similarity.

---

**Algorithm 1** Intrinsic Dimension Estimation Across Layers

**Require:** $n$: number of images; $\Lambda$: number of layers; model$(\cdot, \lambda)$: layer-wise representation
**Ensure:** $ID[\lambda]$: estimated intrinsic dimension at each layer
    $S \leftarrow$ randomly select $n$ images
    **for** $\lambda = 1$ **to** $\Lambda$ **do**
      $Z[\lambda] \leftarrow$ model$(S, \lambda)$
      **for** $i = 1$ **to** $n$ **do**
        Compute $d_1, d_2$: distances to nearest neighbors of $Z[\lambda][i]$
        $R[i] \leftarrow d_1/d_2$
      **end for**
      Estimate $ID[\lambda]$ from the distribution of $R$
    **end for**
    **return** $ID$

---

as well as those that share semantic meaning but have different visual forms. Each of the 10-word pairs consists of a base word (selected from the ToT dataset) and two related words: the **Paronyms Pair**, which refers to words that are visually similar but differ in meaning, and the **Synonyms Pair**, which refers to words that have similar meanings but distinct spellings. All words are real-world entities and are commonly used. For example, as shown in Figure 2, 'goose' is paired with 'moose' as its Paronyms Pair and 'gander' as its Synonyms Pair. This dataset enables a detailed analysis of how models process both visual and semantic similarities in language.

### 3.2. Estimating the Intrinsic Dimensions of ToT Datasets

For images containing varying levels of semantic complexity, we estimate the Intrinsic Dimension (ID) of their representations layer by layer. As Algorithm 1 shows, this process involves using the ID's magnitude as a metric to evaluate how specific layers of the model articulate the textual semantics embedded within the images.

Our study focuses on ViT-based vision models, particularly CLIP ViT-B/16 (cli), given its widespread use in multimodal pretraining. The results on additional architectures are presented in the appendix. CLIP ViT-B/16 is based on a Vision Transformer (ViT) architecture with 12 transformer blocks, and our analysis examines the representations extracted from each block. Within each block, we evaluate three key layers: Attn refers to the output of the attention mechanism after the linear transformation. c_fc is a feedforward expansion layer that projects features from 768 to 3072 dimensions. c_proj is a projection layer that reduces the dimensionality from 3072 back to 768.

The TwoNN algorithm (Facco et al., 2017) estimates the intrinsic dimension (ID) of visual representations by ana-

lyzing the distances between nearest neighbors in a dataset. Algorithm 1 details the procedure applied to layers in a pre-trained model. Specifically, model$(S, \lambda)$ extracts the representation of layer $\lambda$ for the image set $S$, and $ID[\lambda]$ stores the estimated ID values for each layer.

The algorithm computes the distance ratios $R[i] = \frac{d_1}{d_2}$ for each sample, where $d_1$ and $d_2$ are the distances to the first and second nearest neighbors, respectively. For higher intrinsic dimensions, the ratios $R$ follow a Pareto distribution, denoted as $Pa(d + 1)$. This relationship is captured by the likelihood function:

$$P(\mathbf{R}|d) = d^N \prod_{i=1}^{N} R[i]^{-d-1},$$

where $P(\mathbf{R}|d)$ represents the likelihood of observing the distance ratios $\mathbf{R}$ given an intrinsic dimension $d$. A linear fit on the log-transformed distance ratios is then used to estimate the intrinsic dimension, maximizing the likelihood under the assumed Pareto distribution.

## 4. ID Analysis of Textual and Textural Representations in Typographic Attacks

### 4.1. Layer Sensitivity to Typographic Attacks

To analyze how different layers of visual models respond to semantic variations, we first cluster image representations with various textual overlays using t-SNE, offering a preliminary two-dimensional visualization. However, since t-SNE is limited to low-dimensional spaces, we further estimate the intrinsic dimensions (ID) of each layer to evaluate whether intermediate layers preserve semantic distinctions in higher-dimensional spaces.

**Representation Clustering.** We sample image represen-

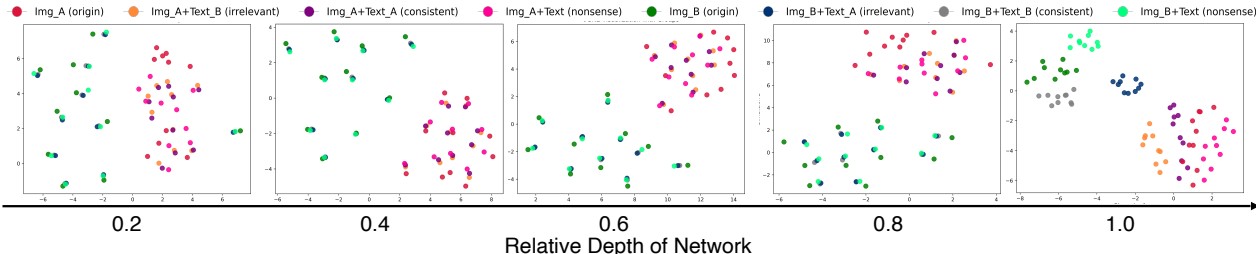

*Figure 3.* t-SNE visualization of representations from ViT-B/16 in different depths. Early layers show two clusters primarily based on image backgrounds, while the final layer shows eight clusters corresponding to all semantic categories.

tations from different network depths and visualize them using t-SNE (Van der Maaten & Hinton, 2008), as shown in Figure 3. In the earlier layers, the representations form two clusters, likely reflecting variations in image background content. In contrast, the final layer reveals eight distinct clusters, capturing a combination of image and text semantics. This pattern suggests that multi-modal visual models initially treat text as a visual texture before gradually incorporating semantic information in deeper layers. To explore this hypothesis further, we estimate the intrinsic dimensions (ID) of representations across layers in the next section.

**Intrinsic Dimensionality Estimation.** We randomly sampled 2,000 images from each subset of the ToT dataset to estimate the Intrinsic Dimension (ID) of representations in the CLIP visual model. The results, shown in Figure 4, reveal a swell-shrink pattern across network layers: representation complexity initially increases before decreasing. This behavior, consistent with prior findings in CNN visual models (Ansuini et al., 2019; Muratore et al., 2022), also emerges in Transformer-based models, aligning with the information bottleneck theory (Shwartz-Ziv & Tishby, 2017), which describes an early fitting phase followed by compression. Notably, while ID values fluctuate across layers, their ratios to the original image's ID remain stable. Typography is observed to consistently increase representational complexity by a factor of 1.2 to 1.3 across most layers.

However, in the final block, nonsensical and irrelevant subsets show significantly higher IDs than the original, while consistent images exhibit a notable decrease. This discrepancy, particularly pronounced in the last layer closest to the classification layer, suggests that the final block has a significant impact on the semantic representation of the entire image. In addition to ViT-B/16, we also tested the IDs of multiple visual encoders with different architecture, the results are shown in Figure 9 in the appendix.

Overall, typography increases the complexity of representations in the intermediate layers, regardless of the semantic relationship between the image and text. However, in the final block, text overlay primarily influences the semantic

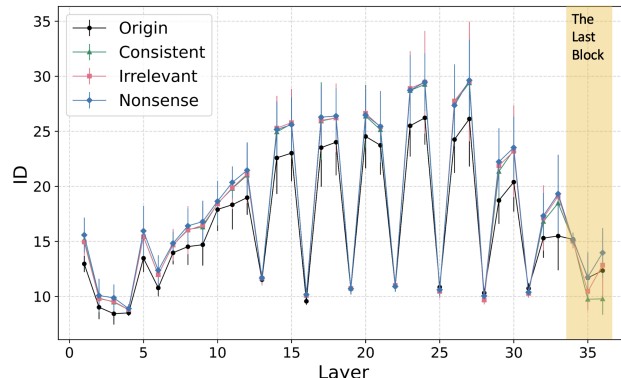

*Figure 4.* The Intrinsic Dimension (ID) variations of ViT-B/16 on the ToT datasets.

aspect of representations. Notably, when the text closely relates to the image content, it appears to reduce representational complexity, as indicated by lower ID values in this layer.

### 4.2. Disentangling Textual and Textural Representations

To better understand the findings in Section 4.1, we design two experiments to disentangle orthographic and semantic representations and analyze their effects on typographic attack. Semantic Constancy using the subset 1 of ToT to examines how variations in font size affect texture-level representations while keeping semantic content unchanged. Linear Probe leverages paronym-synonym pairs (subset 2 of ToT ) to assess the model's progressive disentanglement of textual and textural features across layers. Despite their different implementations, both experiments aim to characterize the interaction between texture and semantics in vision-language models.

**Semantic Constancy with Varying Font Size.** Despite differences in visual appearance, images containing text of varying sizes are often perceived as semantically equivalent (Figure 8, appendix). This raises the question of how visual models encode such variability while preserving se-

*Table 1.* Accuracy (%) on the ToT dataset for multimodal (CLIP, MetaCLIP) and vision-only (ViT, DINOv2) models across different semantic contexts and font sizes. Numeric suffixes (e.g., 80 or 120) indicate font size. Cons, Nons, and Irr refer to consistent, nonsensical, and irrelevant textual content, respectively. Examples are shown in Figure 8 (Appendix).

|          | Orig | Cons_80 | Nons_80 | Irr_20 | Irr_40 | Irr_60 | Irr_80 | Irr_100 | Irr_120 |
|----------|------|---------|---------|--------|--------|--------|--------|---------|---------|
| CLIP     | 86.6 | **98.4** | 80.4   | 78.8   | 60.7   | 49.2   | 42.9   | 40.5    | 38.9    |
| MetaCLIP | 84.7 | **94.6** | 79.3   | 83.3   | 79.6   | 75.2   | 72.1   | 70.0    | 68.4    |
| ViT      | **91.1** | 86.8 | 86.3   | 91.0   | 89.9   | 88.9   | 87.1   | 85.9    | 84.2    |
| DINOv2   | **82.8** | 82.4 | 82.0   | 82.7   | 82.6   | 82.4   | 81.8   | 80.7    | 80.2    |

mantic consistency. We compare multimodal models (CLIP (Radford et al., 2021) and MetaCLIP (Xu et al., 2023)) with unimodal vision models (ViT-B/16 (Dosovitskiy et al., 2022) and DINOv2 (Oquab et al., 2023)), evaluating their performance across variations in text size and semantic context. To ensure a fair comparison, both models share the same ViT-B/16 backbone architecture. The ViT-B/16 model, pretrained on ImageNet-1k (Russakovsky, 2015), serves as the baseline for vision-only models. Both models are evaluated on the ToT dataset, with results presented in Table 1.

Multimodal models exhibit marked sensitivity to both the semantics and the visual appearance of overlaid text. For instance, CLIP achieves 98.4% accuracy in the Cons_80 condition, but its performance drops to 42.9% under Irr_80, despite the font size remaining constant. This suggests that the model's internal representations are shaped not only by surface-level features, such as texture, but also by the semantic alignment between text and image.

In contrast, vision-only models show minimal performance variation across Cons_80, Nons_80, and Irr_80, regardless of semantic content. This contrast indicates that while multimodal models integrate textual meaning into visual understanding, **pure vision models primarily treat text as texture, not as language**.

**Linear Probe on Paronym-Synonym Pairs.** We apply a linear binary classifier probe to the final outputs (ln_2 layer) of all 12 Residual Attention Blocks for both Synonyms and Paronyms pairs. For each pair in subset 2 of the ToT dataset, we use 320 image samples for training and 80 for testing. Following the approach used in CLIP's linear probe experiments, we employ logistic regression as the classifier.

As shown in Figure 5, each lighter-colored line represents a paronym pair (orthographically similar, pink) or a synonym pair (semantically similar, orange), the darker lines indicate the average accuracy of the corresponding 10 pairs. It is evident that all layers achieve higher accuracy when classifying based on orthographic similarity. However, the layers with the steepest slopes for these curves show a distinct pattern: the significant improvement for texture features occurs primarily in the middle layers, whereas the notable enhance-

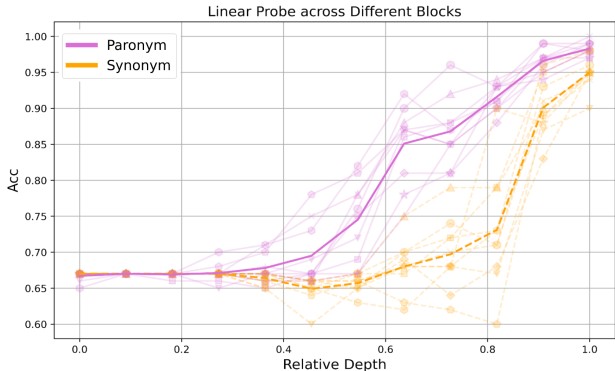

*Figure 5.* Linear probe results: Classification accuracy of each layer for paronym pairs (pink) and synonym pairs (orange). Lighter lines represent individual pairs, while darker lines show the average accuracy for each set of 10 pairs.

ment for textual features is concentrated in the layers closer to the output.

### 4.3. Tracking Semantic Emergence through Intrinsic Dimension

To investigate how visual models progressively form textual representations across layers, we analyze the relationship between intrinsic dimension (ID) and semantic decoding performance, as shown in Figure 6. The combined results reveal a two-phase trajectory—initial representational expansion followed by compression—which aligns with the information bottleneck theory (Saxe et al., 2019). Within this broader pattern, we identify four functionally distinct stages: initialization, texture dominance, compression, and semantic integration.

**Random Initialization.** In the early layers, representational complexity increases gradually, but neither orthographic nor semantic features are reliably formed. Both paronym and synonym classification accuracies remain near chance.

**Texture Dominance.** Intrinsic dimension rises substantially, reflecting increasing representational complexity. Orthographic decoding (i.e., paronym classification) improves rapidly, driven by heightened sensitivity to low-level visual

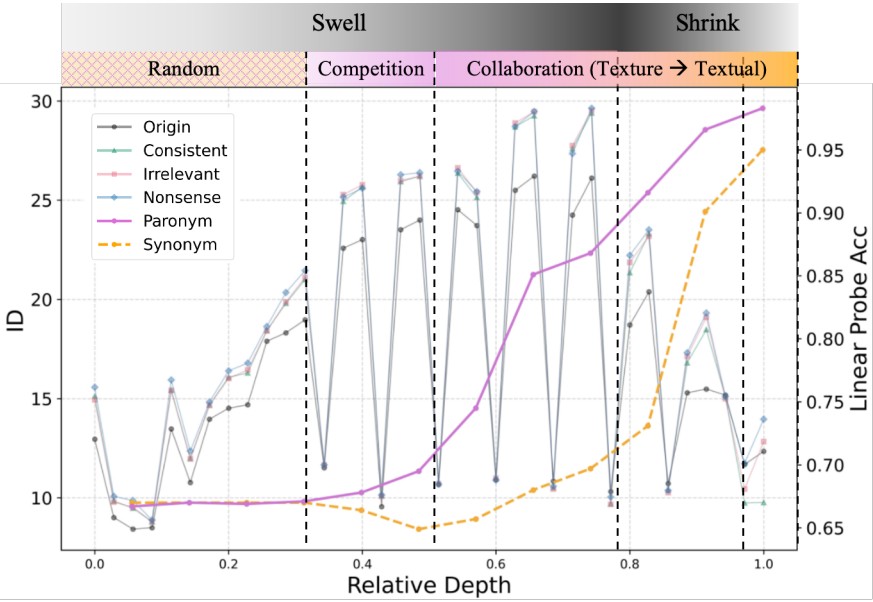

*Figure 6.* Combined visualization of intrinsic dimension (ID) and linear probe accuracy (from Figures 4 and 5), illustrating the progression of text representations across network layers.

features. During this stage, textural and semantic signals initially compete but gradually begin to co-activate within shared representational space.

**Compression.** In the mid-to-late layers, the model undergoes sharp compression, as evidenced by a steep drop in ID. Semantic decoding accuracy (e.g., synonym classification) increases markedly during this period. However, no significant divergence in ID is observed across different semantic categories, suggesting that the learned representations remain largely texture-dependent.

**Semantic Integration.** In the final block, semantic accuracy peaks, and ID values diverge significantly across semantic categories. This indicates that semantic variation is increasingly reflected in the structure of the learned representations.

Together, these observations suggest that while semantic accuracy increases gradually across layers, true semantic differentiation does not emerge progressively. Instead, it appears abruptly at the final block. We further highlight three key findings:

**(1) Overlapping Features in Textual and Textural Encoding.** Across most layers, textural and semantic features rely on partially shared representations. Semantic encoding consistently lags behind, indicating a dependency on lower-level visual patterns during early stages.

**(2) Texture-Induced Accuracy Gains.** Early improvements in semantic decoding may be misleading, as they reflect the model's reliance on texture-based patterns rather than semantic comprehension in its commonly assumed

form.

**(3) Late-Stage Semantic Differentiation.** A clear separation based on semantic content emerges only at the final block, marking a shift from texture-level correlation to semantic abstraction.

# 5. Defense Against Typographic Attacks through Fine-Tuning

Building on the findings in Section 4, different layers of the visual model encode text in distinct ways, influencing its vulnerability to typographic attacks. This suggests a targeted defense strategy: fine-tuning specific layers to enhance robustness. In this section, we explore two approaches, selectively fine-tuning individual blocks (Section 5.1) and fine-tuning only the last block (Section 5.2), to assess their effectiveness in mitigating typographic attacks.

To verify this, we design three typographic attack tasks of varying difficulty: **Easy**: Recognizing image content while ignoring text, similar to the setup in most typography attack work (Materzyńska et al., 2022; Ilharco et al., 2022; Azuma & Matsui, 2023). **Medium**: Detecting the presence of text without understanding its meaning. **Hard**: Distinguishing the semantics of both text and image.

The progression from easy to hard illustrates the increasing complexity of semantic understanding required at each stage. Ideally, fine-tuning only the swell blocks should not effectively defend against any level of attack. In contrast, fine-tuning the shrink and last in shrink blocks should

*Table 2.* Accuracy (%) comparison of defense performance when fine-tuning different parts of the CLIP visual encoder.

| Fine-tuned Blocks | Orig | Easy | | | Medium | | | Hard | | |
|---|---|---|---|---|---|---|---|---|---|---|
| | | Cons | Irr | Nons | Cons | Irr | Nons | Cons | Irr | Nons |
| CLIP w/o ft | 82.3 | 97.3 | 50.6 | 73.7 | 94.5 | 65.9 | 77.4 | 14.5 | 59.9 | 77.2 |
| Swell | 62.8 | 85.8 | 39.1 | 51.2 | 79.0 | 52.1 | 56.3 | 6.6 | 29.7 | 56.2 |
| Shrink | 82.6 | 98.6 | 43.0 | **77.0** | 97.8 | 68.9 | 81.3 | 32.6 | **70.8** | 81.1 |
| Shrink - Last | 83.5 | **98.7** | 32.0 | 76.6 | **98.1** | 61.2 | **81.4** | **36.0** | 68.9 | **81.7** |
| All | 54.8 | 50.1 | 50.3 | 49.1 | 48.8 | 48.9 | 48.1 | 0.1 | 0.5 | 48.2 |
| Last (Ours) | **84.7** | 98.2 | **60.0** | 76.8 | 96.7 | **74.5** | 81.0 | 35.0 | 69.5 | **81.7** |

*Figure 7.* Example image-text pairs from the dataset across easy, medium, and hard defense levels (ground truth).

*Table 3.* Cross-evaluation of SOTA defense methods on handwritten typographic datasets (easy level). Results are reported as accuracy (%).

| Method | Disentangle | PAINT | Prefix | Avg. |
|---|---|---|---|---|
| CLIP | 43.3 | 50.0 | 47.2 | 46.8 |
| Disentangle | **77.8** | 55.5 | 57.6 | 63.6 |
| PAINT | 53.2 | 58.2 | 53.6 | 55.0 |
| Prefix | 71.9 | 63.6 | 58.0 | 64.5 |
| Ours | 73.3 | **68.2** | **67.0** | **69.5** |

provide varying levels of defense based on semantic comprehension. For example, medium difficulty may only require recognition of word orthography, necessitating adjustments to the shrink blocks, while the hard level requires understanding specific meanings, thereby requiring fine-tuning of the last block for effective defense.

All of our experiments are conducted on a GeForce RTX 3090 GPU. We use a batch size of 512 and a learning rate of $1 \times 10^{-4}$, with a weight decay of 0.2. The Adam optimizer is employed for fine-tuning.

## 5.1. Block-Specific Fine-Tuning for Textual and Textural Control

We divide the CLIP encoder into three sections: Swell, Shrink-Last, and Last, as described in Section 4. We fine-tune each section on the hard-level task and evaluate their performance across easy, medium, and hard tasks. The results are shown in Table 2.

Fine-tuning the Swell block alone yields suboptimal performance across all difficulty levels, particularly in tasks requiring semantic understanding. Fine-tuning the Last block proves most effective, particularly in handling higher complexity tasks like Hard-Nons (81.7%) and maintaining high Orig performance (84.7%).

The Shrink strategy also performs well, especially in tasks

requiring nuanced text-image understanding, with strong results in the Medium and Hard categories (70.8% in Hard-Irr). However, fine-tuning the Shrink-Last module provides a balanced performance, almost matching Last in the most difficult tasks while still lagging slightly in simpler cases like Orig (83.5%). This suggests that while Shrink-Last captures some mid-layer texture refinement, it is not as adept at final-stage semantic comprehension as Last alone.

## 5.2. Fine-Tuning the Last Block

### 5.2.1. DEFENSE WITH IGNORING TYPOGRAPHY

**Setup.** Following standard practices, we fine-tune the model on subsets of original and irrelevant images under the easy setting, simulating a training scenario where textual content is treated as noise. To assess real-world performance, we evaluate our method on three publicly available typographic attack datasets: Disentangle (Materzyńska et al., 2022), PAINT (Ilharco et al., 2022), and Prefix (Azuma & Matsui, 2023), which all feature handwritten text overlaid on notepads. We perform cross-dataset evaluations using the respective test sets provided by each method.

**Results.** Table 3 presents the cross-dataset accuracy of various defense methods. Notably, the Prefix approach fine-tunes only the language model, whereas others fine-tune both vision and language components. Our method, which fine-tunes only the vision model, achieves the best overall performance across all datasets, with a slight exception on

*Table 4.* Comparison of defense accuracy (%) between our method and SOTA approaches (CLIP, Prefix (Azuma & Matsui, 2023), and Disentangle (Materzyńska et al., 2022)) on medium and hard difficulty levels, which require semantic understanding of text rather than simply ignoring it, as in the easy-level setting.

| Method | Orig | Medium | | | Hard | | |
|---|---|---|---|---|---|---|---|
| | | Cons | Irr | Nons | Cons | Irr | Nons |
| CLIP | 82.3 | 94.5 | 66.0 | 77.4 | 14.5 | 59.9 | 77.2 |
| Prefix | 82.0 | 91.4 | 69.9 | 76.0 | 10.6 | 27.1 | 75.5 |
| Disentangle | 79.9 | 85.0 | 72.0 | 75.2 | 12.9 | 13.8 | 75.3 |
| Ours_Med | 83.5 | 92.2 | **82.4** | **82.9** | 8.1 | 22.0 | **82.4** |
| Ours_Hard | **84.7** | **96.7** | 74.5 | 81.0 | **35.0** | **69.5** | 81.7 |

the Disentangle dataset, where it is marginally outperformed by the original method. This cross-evaluation demonstrates that our method is not overly fitted to our ToT dataset but instead captures a generalizable defense strategy effective across diverse typographic attack scenarios.

### 5.2.2. DEFENSE WITH PRESERVING THE TYPOGRAPHY SEMANTICS

Table 4 presents the evaluation results for medium and hard levels of defense, which require the recognition of the absence of words and specific semantics, respectively. Our method outperforms other models across all difficulty levels. The Prefix and Disentangle methods, trained on datasets similar to those used for easy-level tasks, reveal limitations in recognizing character forms and semantics, as demonstrated by their performance in the hard-level results. In contrast, our model exhibits superior comprehension across various difficulty levels, particularly when the image-text relationship is semantically consistent.

Training on datasets with higher difficulty levels presents challenges in balancing 'Cons' and 'Irr' image-text pairings in the medium scenarios. However, in hard scenarios, where understanding both textual and visual semantics is essential, performance can be improved simultaneously. With the appropriate training data, our method effectively fine-tunes models to comprehend both textual and visual semantics.

Another advantage of our approach is its ability to balance adversarial tasks with the original task. As shown in the 'orig' column of Table 4, our methods outperform all other models, despite primarily being trained on typographic samples. Notably, the 'Ours_Hard' model demonstrates improved 'orig' accuracy, even when typographic semantics potentially conflict with original image classification.

## 6. Conclusion

We explore how visual models process textual semantics in the context of typographic attacks. By introducing the ToT dataset and applying Intrinsic Dimension (ID) analysis,

we reveal that early layers of visual models primarily rely on texture features rather than true semantic understanding. Only in the final block do models construct a meaning-driven semantically understanding after significant compression of textural information. Furthermore, we demonstrate an effective defense strategy by fine-tuning the final block, which enhances the model's ability to distinguish between textural and textual elements. This approach significantly improves performance across various defense scenarios, offering a practical solution to typographic attacks.

## Acknowledgements

This work was supported by the National Natural Science Foundation of China (62206167, 82430060).

## Impact Statement

This paper investigates how multimodal vision models process textual semantics, revealing that early layers rely more on visual textures than on textual meaning, which makes these models vulnerable to typographic attacks.

By shedding light on the internal mechanisms of these models, this work advances the transparency and accountability of machine learning systems. It contributes to the broader effort to make AI more interpretable and aligned with societal values, particularly as these systems are increasingly deployed in decision-making processes. From a societal perspective, the vulnerabilities exposed in this research have important implications for the safety and fairness of AI systems in real-world applications. Typographic attacks could lead to unintended consequences in fields such as law enforcement, healthcare, or finance, where misinterpretations could disproportionately affect vulnerable groups. Ensuring that models are both transparent and secure is crucial for their responsible integration into these high-impact areas.

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

# A. Appendix

## A.1. Details of the ToT Datasets

To create the Textural or Textual (ToT) dataset, we follow the approach of PAINT and Prefix. We resize the images to 224 pixels in the shorter dimension using bicubic interpolation and crop a 224x224 pixel area from the center, consistent with standard CLIP resizing and cropping techniques. The text is randomly overlaid at arbitrary positions on the images.

**Font.** We randomly select from Roman, Courier, and Times fonts and utilize eight colors: black, blue, cyan, green, magenta, red, white, and yellow. The text is outlined with a 1-point shadow in a contrasting color.

**Font sizes.** We use 80 points to generate images for the Consistent, Irrelevant, and Nonsense categories. Additionally, to further investigate the impact of font size on identification (ID), Irrelevant images are created in font sizes ranging from 20 to 120 points. The examples are shown in Figure 8.

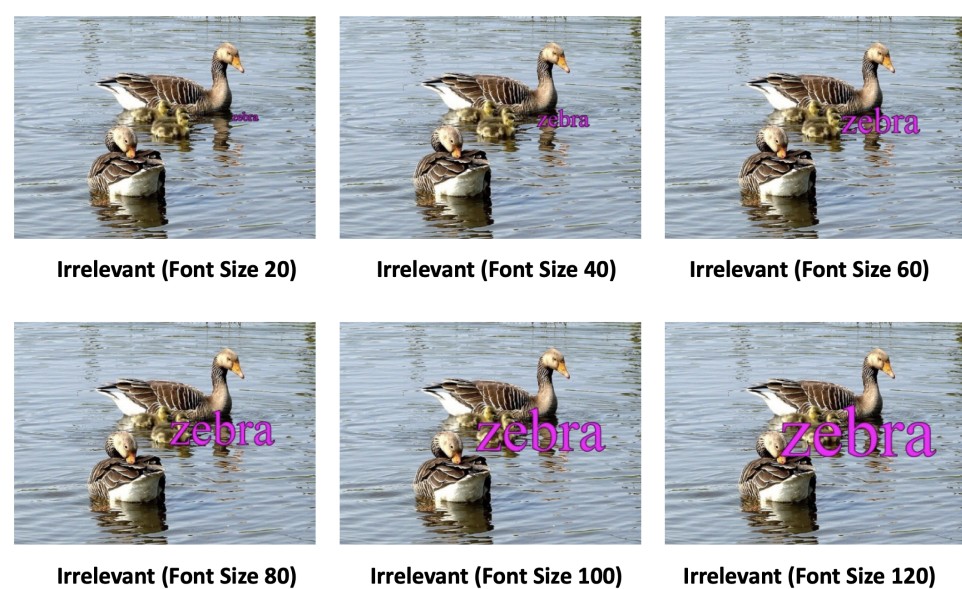

*Figure 8.* Examples of typography with different sizes.

**Subset 1 Categories.** The 100 categories of the ToT datasets are peacock, goose, koala, jellyfish, snail, flamingo, sea lion, Chihuahua, tabby cat, lion, tiger, bee, dragonfly, zebra, pig, llama, panda, backpack, barrel, basketball, bikini, bottlecap, bow, broom, bucket, buckle, candle, cannon, cardigan, carton, coffee mug, coffeepot, crib, envelope, fountain, iPod, iron, jean, ladle, laptop, lighter, lipstick, lotion, mailbox, mask, microwave, mitten, mouse, nail, necklace, paddle, pajama, perfume, pillow, plastic bag, printer, projector, purse, radio, refrigerator, ruler, shovel, sock, stove, suit, sunglass, swing, switch, table lamp, teapot, television, toaster, tray, tub, umbrella, vacuum, vase, violin, wallet, whistle, ice cream, bagel, hotdog, cucumber, mushroom, Granny Smith, strawberry, orange, lemon, banana, hay, dough, pizza, potpie, red wine, espresso, cup, volcano, daisy, and corn.

**Subset 2 Categories.** The subset includes the following paronym-synonym pairs: Goose (n01855672): Moose, Gander; Bee (n02206856): Beef, Wasp; Pig (n02395406): Fig, Hog; Fountain (n03388043): Mountain, Spring; Mitten (n03775071): Kitten, Glove; Nail (n03804744): Mail, Spike; Hay (n07802026): Ray, Straw; Espresso (n07920052): Express, Coffee; Lemon (n07749582): Demon, Lime.

## A.2. ID Variations of Different Architectures and Datasets

**Other vision models.** To further assess the generalizability of our findings, we examined the Intrinsic Dimension (ID) of visual encoders from several multimodal architectures, beyond the ViT-B/16 used in CLIP. Figure 9 shows the intrinsic dimension curves of these models. Except for DINOv2, which is a vision-only pretrained model, all other models exhibit a clear divergence in ID at the final block when different semantic texts are overlaid.

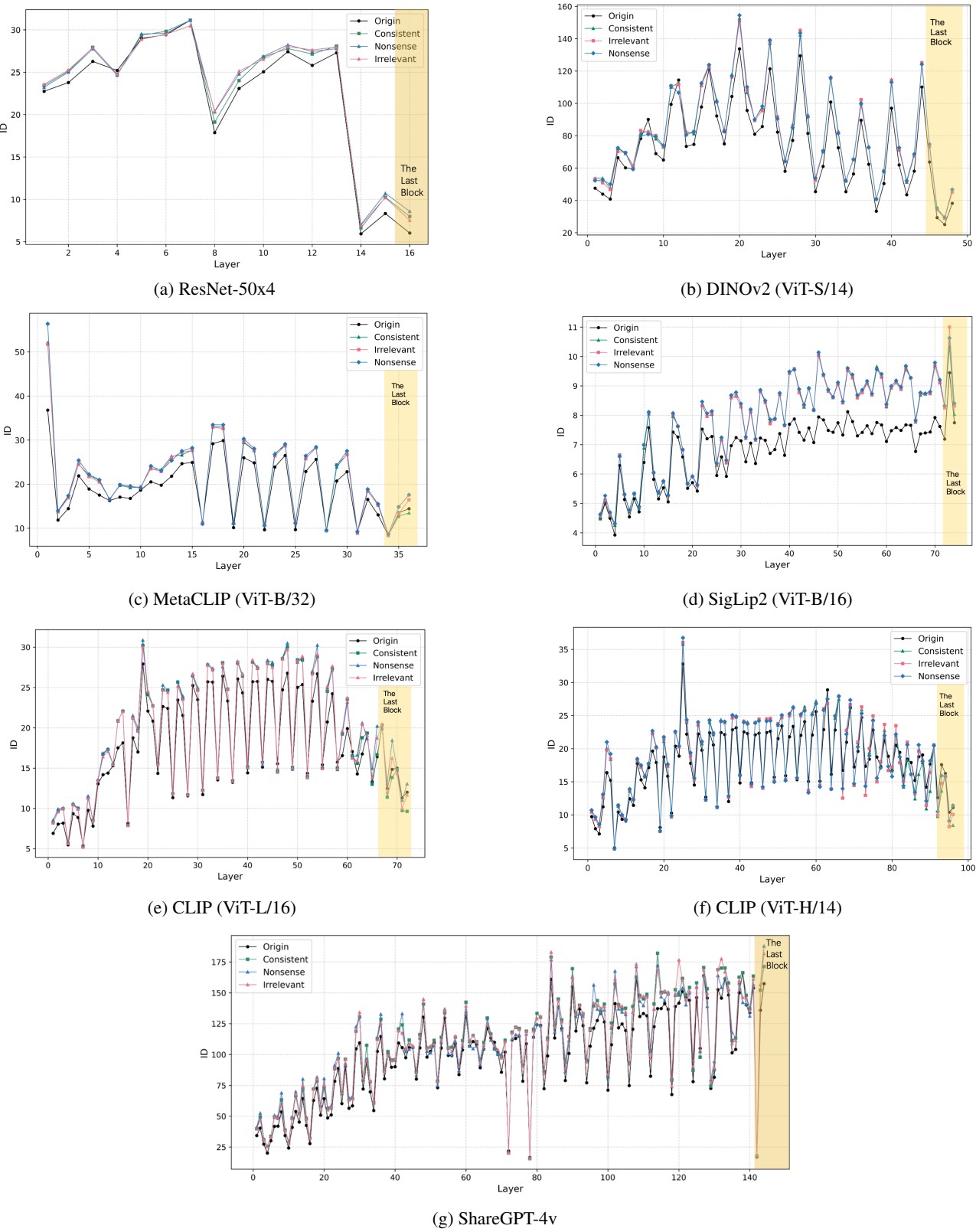

(a) ResNet-50x4

(b) DINOv2 (ViT-S/14)

(c) MetaCLIP (ViT-B/32)

(d) SigLip2 (ViT-B/16)

(e) CLIP (ViT-L/16)

(f) CLIP (ViT-H/14)

(g) ShareGPT-4v

*Figure 9.* Layer-wise intrinsic dimensions (IDs) of visual encoders across different architectures.

**Other datasets.** While our primary focus is on the ToT dataset, we observe that the behavior of the Intrinsic Dimension (ID) also holds across other datasets. Using the same methodology as for the ToT dataset, we applied text overlays to the Caltech101 dataset. As shown in Figure 10, the resulting ID variations follow the similar trend as those observed with the ToT dataset.

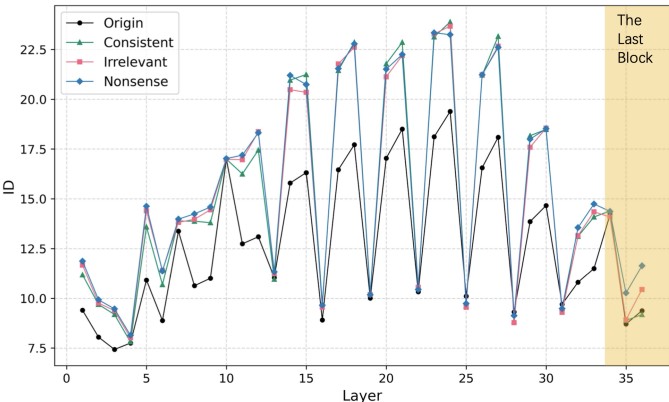

*Figure 10.* IDs of ViT-B/16 on the Caltech101 Dataset.

### A.3. Defense Performance on MetaCLIP

*Table 5.* Performance comparison when fine-tuning different parts of the MetaCLIP visual encoder.

| Fine-tuned Blocks | Orig | Easy | | | Medium | | | Hard | | |
|---|---|---|---|---|---|---|---|---|---|---|
| | | Cons | Irr | Nons | Cons | Irr | Nons | Cons | Irr | Nons |
| Swell | 19.3 | 17.5 | 16.7 | 15.7 | 17.1 | 16.8 | 15.9 | 0.4 | 0.1 | 16.0 |
| Shrink | 70.5 | 89.3 | 63.2 | 65.0 | 88.0 | 60.7 | 63.3 | 17.5 | 46.3 | 62.5 |
| Shrink - Last | 73.1 | 87.5 | 64.8 | 66.4 | 86.7 | 62.4 | 65.1 | 22.8 | 44.8 | 64.0 |
| Last (Ours) | **83.6** | **91.3** | **79.0** | **80.0** | **90.3** | **77.4** | **78.2** | **40.3** | **54.7** | **78.3** |

### A.4. Correlation Between ID and Accuracy

Table 6 shows the relationship between the ID of the last fully connected layer (ID_Last), the maximum ID (ID_Max), and classification accuracy for ViT and CLIP models. The Spearman correlation coefficient (Spearman, 1987) is used to measure the correlation between accuracy and ID values.

The correlations for ViT are $\rho(\text{ID\_Last}, \text{Acc}) = -0.98$ and $\rho(\text{ID\_Max}, \text{Acc}) = -0.63$, while for CLIP, they are $\rho(\text{ID\_Last}, \text{Acc}) = -0.13$ and $\rho(\text{ID\_Max}, \text{Acc}) = -0.73$. Overall, an inverse correlation is observed, suggesting that lower ID values correspond to higher classification accuracy. However, this trend is not consistent for the last layer ID, deviating from patterns typically found in standard image classification tasks (Ansuini et al., 2019).

*Table 6.* Correlation between classification accuracy and ID values for the last layer and maximum ID across layers, differentiated by typography type and size. The models compared are pre-trained via multimodal (CLIP) and pure vision (ViT) models.

| Model | | Orig | Cons_80 | Nons_80 | Irr_20 | Irr_40 | Irr_60 | Irr_80 | Irr_100 | Irr_120 |
|---|---|---|---|---|---|---|---|---|---|---|
| ViT | Acc. | 91.1 | 86.8 | 86.3 | 91.0 | 89.9 | 88.9 | 87.1 | 85.9 | 84.2 |
| | ID_Last | 6.8 | 7.8 | 8.0 | 7.0 | 7.2 | 7.6 | 7.8 | 8.1 | 8.3 |
| | ID_Max | 89.9 | 92.4 | 95.4 | 95.6 | 89.1 | 95.2 | 100.0 | 99.1 | 102.5 |
| CLIP | Acc. | 86.6 | 98.4 | 80.4 | 78.8 | 60.7 | 49.2 | 42.9 | 40.5 | 38.9 |
| | ID_Last | 12.4 | 9.8 | 14.0 | 14.8 | 14.3 | 13.0 | 12.7 | 12.5 | 12.6 |
| | ID_Max | 26.2 | 29.4 | 29.6 | 29.2 | 29.5 | 29.4 | 29.5 | 29.8 | 29.8 |

For the ViT model, there is a clear correlation between accuracy and text size: as text size increases, accuracy decreases,

which aligns with changes in ID_last values. This observation supports findings from (Ansuini et al., 2019), where text size plays a significant textural role in pure vision models.

In contrast, the CLIP model shows that text semantics significantly impact accuracy, even when size is controlled. The relationship between accuracy and ID metrics is more complex here; while no clear correlation exists with ID_last for semantically irrelevant texts, there is a strong inverse correlation between accuracy and ID_max as text size increases. This suggests that ID_max captures textural complexity, whereas ID_last reflects both textural and textual features. CLIP's representation of text involves a complex interaction between these elements, with semantics heavily influencing accuracy, yet no single layer fully captures this correlation.

