# OpenReview forum: "Textural or Textual: How Vision-Language Models Read Text in Images"
_ICML.cc/2025/Conference — ICML 2025 poster_

### Official Review · Reviewer_HHNW · 2025-03-06

**Overall Recommendation:** 2

**Summary:**

This paper systematically analyzes how the encoder-only vision-language models (i.e., CLIP) perceives the textual and semantic information. The paper uses ID and linear probe to measure the representation complexity and semantic perception ability of vision-language models. The analysis suggests that at earlier layers the model captures texture information while in the later layers, the model captures semantic information. Based on above findings, the paper proposes a method for against typographic attacks by solely fine-tuning the last layer and the experimental results show the improved performance.


## update after rebuttal
Thanks for the effort, since the concerns are not well addressed, I decide to keep the score.

**Claims And Evidence:**

The two main experiments designed in Section 4.2 can not fully support the claims.

1. Semantic Constancy with Varying Font Size: The author hypothesizes that increasing font size enhances texture complexity, thereby influencing the model's judgment. However, changes in font size may simultaneously affect semantic readability (e.g., smaller fonts may be harder to recognize), introducing confounding variables. For instance, the ViT model's accuracy decreases with larger fonts (Table 1), which the author attributes to texture interference. However, it is possible that the actual reason is that larger fonts obscure more image content, rather than solely due to changes in texture.

2. ​Linear Probe on Paronym-Synonym Pairs​: The author conduct experiments on only 10 pairs of words, which constitutes a relatively small sample size. Furthermore, is the selection of word pairs balanced in terms of factors such as word frequency and visual similarity? For instance, do the visual and semantic differences between "goose-moose" and "goose-gander" adequately represent the broader spectrum of such comparisons? If there is bias in the selection of word pairs, the conclusions drawn may lack generalizability.

The conclusion drawn from observations are problematic:

1. Figure 5 shows a significant improvement in the classification accuracy of synonyms in deeper layers, which the authors attribute to the delayed formation of semantic understanding. However, the increase in accuracy might merely reflect that deeper features are more linearly separable, rather than indicating the enhanced comprehension of semantic. For instance, deeper features could become easier to classify due to dimensionality compression, but this does not necessarily imply semantic encoding.

2. The authors interpret the decrease in ID as semantic compression. However, the reduction in ID could also stem from decreased feature redundancy or noise suppression, which may not directly correspond to semantic abstraction. Additional evidence, such as feature visualization or intervention experiments, is needed to substantiate the causal relationship between changes in ID and semantic representation.

**Essential References Not Discussed:**

No

**Experimental Designs Or Analyses:**

The experiment presented in Section 5.1 needs another baseline, i.e., fine-tuned the whole layers.

**Methods And Evaluation Criteria:**

See claims and evidence

**Other Comments Or Suggestions:**

See claims and evidence.

**Other Strengths And Weaknesses:**

Strengths: The paper is well-structured.

Weakness: see claims and evidence.

**Questions For Authors:**

No question

**Relation To Broader Scientific Literature:**

This paper engages with the existing literature in the following aspects:

It proposes a defense strategy by solely fine-tuning the last layer, balancing efficiency and performance.

**Theoretical Claims:**

NA

---

> ### Author Rebuttal · Authors · 2025-03-31
>
> We thank the reviewer for the detailed comments. We appreciate the focus on the underlying assumptions behind our interpretation, and we have carefully examined all concerns related to experimental validity and inference logic. We address each point below, and will incorporate clarifications, results, and new visualizations into the revised version. Additional materials can be found at: https://anonymous.4open.science/r/TOT-078C/
>
> ---
>
> **Q1. Font size may introduce multiple confounds (texture vs. visibility vs. occlusion)**
>
> We agree that font size may introduce occlusion or legibility effects. To control for this, we included a **vision-only ViT as a baseline** in Table 1. Since ViT lacks language input, its accuracy drop reflects purely visual factors.
>
> CLIP, with the same visual backbone, shows a different trend, performance degrades more when the overlaid text **conflicts semantically** with the image. This contrast suggests that CLIP’s behavior cannot be explained by occlusion alone, and that semantic interference plays a key role.
>
> **Q2. Small sample size in paronym-synonym probing; representativeness of word pairs**
>
> The probe experiment was designed as a controlled interpretability analysis using **representative and well-matched word pairs**. Each pair was selected to reflect either semantic or orthographic similarity, while controlling for **word frequency, visual appearance, and word length**. All words used are **frequent and familiar**, ensuring they are easily interpretable by both humans and models.
>
> While the total number of pairs is small, they were chosen to **typify common semantic vs. paronym contrasts** (e.g., “goose–gander” vs. “goose–moose”). The key observation holds consistently across all pairs, supporting the robustness of the trend.
>
> We will clarify the selection criteria and framing in the revision, and agree that expanding to a larger lexical set is a valuable direction for future work.
>
> **Q3. Increased accuracy in deep layers may not imply semantic abstraction**
>
> We understand the concern that deeper layers may naturally yield higher separability. However, **if this were purely due to compression or feature refinement,both paronym and synonym distinctions should improve similarly**.
>
> Instead, we observe a clear asymmetry: orthographic separability is strong from early layers, while semantic separability emerges only in the final block. This pattern cannot be explained by general separability alone, and provides a strong structural signal of **delayed semantic abstraction**.
>
> **Q4. ID drop may reflect redundancy reduction, not semantic compression**
>
> We agree that a decrease in ID can result from various factors such as redundancy reduction or noise suppression, and does not directly prove semantic abstraction. In our work, we treat ID as a proxy for representational complexity, not for semantic content itself.
>
> To support our interpretation, we provide **multiple converging signals**:
>
> 1. ID Analysis shows a **consistent drop** in the final block, particularly under semantic perturbations (e.g., synonym substitutions), suggesting a shift in representational dynamics.
> 2. Linear Probing reveals that only the final block supports linear separation of semantic distractors (synonyms), whereas paronym separability appears **much earlier**.
> 3. **Grad-CAM visualizations** (Figure C1) show that:  in the final block, **attention re-centers on the object only when the overlaid text is semantically aligned with the image**. In contrast, for irrelevant or nonsensical text, the model continues to attend to the text region, even in the deepest layers.
>
> Together, these findings suggest that semantic abstraction is not merely a byproduct of general compression, but emerges **selectively and meaningfully** in the representational structure. We do not claim causal proof, but this triangulation offers strong evidence of **a structured link between ID reduction and semantic resolution**.
>
> We will clarify this framing and add more explanation for Figure C1 in the revision.
>
> **Q5. Lack of full-model fine-tuning baseline in Section 5.1**
>
> Full-model fine-tuning (**Table A1**) underperforms our final-block-only strategy by **over 30%** across all splits, likely due to **overfitting and disruption of early representations**. Our analysis shows that semantic abstraction emerges in the final block, making targeted fine-tuning both more effective and efficient.
>
> ---
> Thank you for your thoughtful and precise feedback. We’ve carefully responded to the points raised to clarify assumptions, tighten the experimental framing, and better reflect the structure of our findings. We hope our explanations help strengthen the rigor and improve the clarity of the work.

---

### Official Review · Reviewer_UJSz · 2025-03-10

**Overall Recommendation:** 3

**Summary:**

This paper investigates images with overlaid text, and how vision-language models process them. The paper analyzes representations throughout model layers using Intrinsic Dimension as a measure of complexity. It finds that early layers primarily encode textures while the last one encodes semantics. Through these insights, they are able to achieve significant improvements against typographic attacks by only fine-tuning the final layer.

**Claims And Evidence:**

- Claim 1. In early layers texture and semantics compete, while in late layers semantic accuracy improves.
    - Sec 3 discusses the design of the evaluation set, which contains paronyms (which are texturally related) and synonyms (which are semantically related).
    - Figure 4 shows that any typography increases ID in middle layers. In the last layer, there is a clear ordering where the most to least consistent typography has the smallest to biggest ID.
    - Table 1 shows for a pure vision ViT is less sensitive to text semantics and more to size whereas multimodal CLIP is heavily sensitive to semantics.
    - I don’t understand the result in Figure 5. It seems to imply that the model achieves higher accuracy on the paronym pairs, for example “moose” overlaid on a picture of a goose, which can also be considering to be a conflicting or “irrelevant” case. This result conflicts with Table 1, where “Cons_80” significantly outperforms “Irr_80” for CLIP. The setup of this experiment is also unclear: is every image duplicated four times, as shown in Figure 2, with different text overlaid, and the classification accuracy is computed averaged across all these duplicated images?
- Claim 2. Building on these insights, fine-tuning only the final block of the model is sufficient to achieve state-of-the-art performance for ignoring typography.
    - This result is supported by Tables 3 and 4. The prior work used as baselines focus on subspace discovery, weight interpolation, or prefix tuning, rather than direct fine-tuning as investigated by this work.

**Essential References Not Discussed:**

N/A

**Experimental Designs Or Analyses:**

See “Claims and Evidence” above.

**Methods And Evaluation Criteria:**

See “Claims and Evidence” above.

**Other Comments Or Suggestions:**

Below are minor clarity and writing comments that do not affect my score, but are intended as constructive feedback.

- I don’t understand L044-045 of the Introduction; is there a prior work that shows that “textual elements […] are often encoded similarly”?
- Typo, in L063 “We” should be lowercased as “we”
- I took me a long time to understand that Figure 6 was a combination of Figures 4 and 5; you should also reference these figures in the figure caption. It would be much easier to read if Figures 4 and 5 were stacked on the same page or in a single figure, rather than overlaid in Figure 6.
- Typo, in L064 the citation for Cao et. al. should use \citet
- In L157, I would recommend renaming the categories [“Consistent”, “Irrelevant”] → [“Matched”, “Mismatched”] or some more symmetric naming.
- In Figure 5 I would recommend renaming [”Orthographic”, “Semantic”] → [”Paronym”, “Synonym”] so it’s easier to understand what is being evaluated.

**Other Strengths And Weaknesses:**

Strengths

- I liked how the insight from Figure 4, where it is identified that most of the semantic separation occurs in the last block, is used to motivate the experiments in Sec. 5.

Weaknesses

- I don’t understand the experiment in Figure 5 / L313, where the result seems to conflict with the result in Table 1 (see “Claims and Evidence” above).

Overall, I liked the premise of investigating texture versus semantics in typographic attacks, and the experiments reasonably supported this investigation. The result in Sec. 5 is also strong. However, the presentation is hard to parse, in particular keeping track of all the evaluation sets described in Sec 3.1.

**Questions For Authors:**

N/A

**Relation To Broader Scientific Literature:**

This work conducts a controlled evaluation of texture versus semantics in typographic attacks (Figure 4, Table 1), and shows that direct fine-tuning of the final block is sufficient to achieve state-of-the-art results (Table 3), compared with more hand-crafted methods in prior work.

**Theoretical Claims:**

N/A

---

> ### Author Rebuttal · Authors · 2025-03-31
>
> Thank you for the thoughtful and encouraging review. We're glad the main idea came through clearly: we aim to understand how vision-language models handle overlaid text, and how representational insights can guide robustness improvements. Your suggestions on Figure 5 and the evaluation setup are especially helpful. Below we respond point by point. Additional materials can be found at: https://anonymous.4open.science/r/TOT-078C/
>
> ---
>
> **Q1. Figure 5 appears to contradict Table 1, and the setup is unclear**
>
> These two experiments address different questions and use different evaluation metrics:
>
> - Figure 5 uses a **probing classifier** to test whether representations at **each layer** contain enough information to distinguish orthographic (paronym) vs. semantic (synonym) distractors.
> - Table 1 evaluates end-to-end **image-text matching accuracy (by the last layer)** under typographic attacks, focusing on task-level performance.
>
> The confusion may stem from the similar word pairs appearing in both contexts. In Table 1, they’re assessed for semantic alignment; in Figure 5, they are diagnostic probes for how semantic and visual cues are encoded across layers. The lower probe accuracy on synonym distractors (e.g., goose-gander) indicates that semantic information is harder to linearly extract at intermediate layers, whereas orthographic differences remain easier to capture. We also provide a detailed explanation of both the evaluation and training setups in **Figures B1 and B2** in the appendix. We hope this helps clarify how the probe task (Figure 5) differs from the classification task (Table 1).
>
> Rather than contradicting, this result **reinforces our central finding**: semantic abstraction emerges later than visual/textural cues in VLMs. We will revise the text to clarify the goals and setup of the probing experiment.
>
> **Q2. L044–045 of the Introduction: Is there prior work showing that "textual elements [...] are often encoded similarly"?**
>
> Thank you for pointing this out. That sentence was intended as a motivating question rather than a factual claim. We agree that the phrase “are often encoded similarly” may overstate the point. In the final version, we will cite prior work that offers perspectives on both sides of this question, and present it with an objective and open stance.
>
> **Q3. Minor typos and naming / layout suggestions**
>
> Thank you for these valuable presentation suggestions. We will revise the manuscript accordingly:
>
> - Clarify that **Figure 6 combines Figures 4 and 5**, and update the caption;
> - Stack Figures 4 and 5 on the same page for easier visual comparison;
> - Rename categories from **"Consistent / Irrelevant" → "Matched / Mismatched"**, and **"Orthographic / Semantic" → "Paronym / Synonym"**, to improve interpretability.
>
> ---
>
> We appreciate your engagement with both the technical and presentation aspects of this work. Your suggestions helped us clarify our framing and improve the accessibility of the paper. We believe the revised version reflects these improvements and presents a clearer, more complete contribution.

---

> > ### Comment · Reviewer_UJSz · 2025-04-01
> >
> > Thank you for clarifying the experiments in Figure 5 and Table 1, and providing additional visualization of the setup in Figures B1 and B2. The differences make sense given that Figure 5 is measuring the classifier's ability to detect orthographic vs semantic differences (where it makes sense that it is overall easier to detect orthographic differences) whereas Table 1 is measuring image-text matching accuracy. The results are indeed consistent and provide a comprehensive picture of the paper's claims.
> >
> > The experiments are thorough and well-motivated. I appreciate the authors' efforts to incorporate my presentation suggestions; however, the clarity of the original submission still impacts my score.
> >
> > I maintain my positive score and recommend the paper for acceptance.

---

> > > ### Author Response · Authors · 2025-04-02
> > >
> > > Thank you for your thoughtful feedback and kind recommendation. We truly appreciate your recognition of our experimental clarifications and your helpful suggestions. We will make sure to improve the clarity of our writing in the final version.

---

### Official Review · Reviewer_8NZn · 2025-03-13

**Overall Recommendation:** 2

**Summary:**

This paper investigates Typographic attacks in vision language models. They investigate whether these models encode textual semantics through representation complexity, and identify the mechanisms by which text disrupts visual understanding.
To decouple orthography from semantics, they introduce the ToT dataset, containing minimal pairs of words that either share semantics with distinct visual forms (synonyms) or match visual forms with conflicting semantics (paronyms).
By analyzing layer-wise Intrinsic Dimension (ID), they found that early layers exhibit competing dynamics between orthographic features and semantics, while later layers improve semantic accuracy primarily through orthographic disambiguation. Crucially, semantically driven representations emerge only in the final block, challenging the assumption of progressive semantic understanding.

## update after rebuttal

Due to this limited scope of experiment, and that the writing of this paper definitely needs to be improved, I'd like to remain my original score.

**Claims And Evidence:**

First of all I'm not familiar with this area of Typographic attacks in vision language models.
After reading the paper, I think most claims are supported with evidence. But there are some confusing parts.
1. It seems that all the conclusions can only apply to CLIP, instead of general vision language models. Could you extend to other models, e.g.DINOv2, SigLIP, MetaCLIP...?
2. In Table 1, how to read the numbers in the first row? It's hard to understand what the authors want to say about the Irr_s.
3. Also,  Sec 3.2 mentions PARONYMS VS. SYNONYMS CONFUSION but their performance comparison is not presented in Table1? Where can we read their experiment results?
4. The Table 3 row names and column names are the same, it's not clear what the "sota methods" are exactly.
5. In Table 2 and 4, under "Hard", why are the Irr and Nons scores much higher than Cons score, opposite to the other columns? Shouldn't Cons always be easier than Irr and Nons?

**Essential References Not Discussed:**

N/A

**Experimental Designs Or Analyses:**

There are some confusing points. See above.

**Methods And Evaluation Criteria:**

There are some confusing points. See above.

**Other Comments Or Suggestions:**

N/A

**Other Strengths And Weaknesses:**

I think the presentation of the paper can be further improved. Also, the finetuning defense method seems a bit lack of novelty. More models and more explanations should be added.

**Questions For Authors:**

See above. Happy to raise my score if questions are addressed.

**Relation To Broader Scientific Literature:**

It's related to safety issues for general vision language models.

**Theoretical Claims:**

N/A

---

> ### Author Rebuttal · Authors · 2025-03-31
>
> Thank you for your thoughtful and open review. This paper is primarily an **interpretability-driven study**: our goal is to understand how VLMs represent and process text in images across layers. Rather than proposing a new defense method, our core contribution lies in using Intrinsic Dimension (ID) to **trace where semantic abstraction emerges, and how it breaks** under typographic attacks. The fine-tuning experiment serves to validate our explanation in a practical setting.
>
> We address your concerns point by point below. Additional experimental results can be found at: https://anonymous.4open.science/r/TOT-078C/
>
> ---
>
> **Q1.  Could you extend to other models (DINOv2, SigLIP, MetaCLIP...)?**
>
> Yes, while our main experiments focus on CLIP (a widely-used baseline), we have extended our analysis to **ViT-L/14, ViT-H/14 DINOv2, SigLIP, and MetaCLIP**. Please see Figures A1–A5 in the supplementary material.
>
> We observe a **consistent semantic abstraction pattern across these models**: semantic distinctions between consistent, irrelevant, and nonsense text inputs remain indistinguishable in early layers, but diverge only in the final block (reflected by their ID separation). This suggests that delayed semantic resolution is a **shared property of VLMs, not limited to CLIP**.
>
> Notably, **DINOv2, a pure vision model** without language pretraining (similar to the ViT in Table 1), does not exhibit such ID bifurcation in the final block, also supporting our interpretation.
>
> **Q2. In Table 1, how should we read the “Irr_s” numbers in the first row?**
>
> The first-row entries like “Irr_80” refer to **irrelevant text overlaid at a font size of 80**. Appendix Figure 8 shows examples. Font size modulates the visual salience of the text: larger fonts induce stronger texture-level interference, allowing us to evaluate robustness across texture intensity.
>
> We will make this explicit in the table caption.
>
> **Q3. Section 3.2 mentions paronym vs. synonym confusion, but their comparison is not in Table 1. Where is it?**
>
> This analysis is in **Figure 5**, via a linear probing experiment across residual blocks.
>
> We trained logistic regression classifiers to distinguish **paronym pairs** (orthographically similar but semantically different) from **synonym pairs** (semantically similar but orthographically distinct). Each line represents a word pair; darker lines are averages over 10 pairs.
>
> This reveals **how orthographic and semantic information evolve across depth**, and aligns with our main claim: semantic abstraction is delayed and fragile.
>
> We will revise Section 3.2 to better connect with this figure and clarify the setup.
>
> **Q4. In Table 3, row and column names are the same. What exactly are the “SOTA methods”?**
>
> Thank you for pointing this out, we see how it could be confusing. Table 3 is a **cross-dataset evaluation**: each row represents a defense method, and each column is the dataset it was trained on (each paper proposes a new dataset). The diagonal shows **in-domain robustness**, while off-diagonal cells reveal **cross-domain generalization**.
>
> We deliberately adopted this structure to **decouple the effects of methods and datasets**, ensuring a more rigorous and fair comparison. Our method achieves the **best or second-best performance across all datasets**, demonstrating its broader applicability. We will clarify this in the table caption.
>
> **Q5. In Tables 2 and 4, under “Hard,” why are Irr and Nons scores higher than Cons? Shouldn't Cons be easier?**
>
> This is a subtle but important point. In the “Hard” setting, the training set contains more Irr and Nons samples than Cons, introducing both a distributional shift and higher conflict complexity (see Fig. 7). As a result, models trained in this setting tend to perform better on the more frequent categories, even if Cons is conceptually easier.
>
> While Cons is generally easier for the **original CLIP**, once we fine-tune the model, it begins to **balance trade-offs across tasks** such as consistent vs. irrelevant recognition. This can lead to non-monotonic changes in relative difficulty, where Cons is no longer always the easiest class.
>
> **Q6. The writing can be improved. The fine-tuning method seems to lack novelty.**
>
> We will revise the paper for clarity, especially in table structure, dataset presentation, and experimental details.
>
> As for the fine-tuning method, it is not positioned as a core novelty, but rather as a **minimal but effective validation** of our main claim regarding layer-wise semantic abstraction. The method is **intentionally simple**: our goal is to show that **understanding** where semantic features emerge allows even lightweight interventions to yield significant robustness gains. This underscores the **practical value** of our interpretability-driven analysis.
>
> ---
>
> We appreciate your thoughtful engagement. We hope our responses have addressed your concerns and clarified our motivation. Thank you for considering a re-evaluation.

---

> > ### Comment · Reviewer_8NZn · 2025-04-06
> >
> > Thank you for your detailed response! Many verification questions become much clearer with the explanations.
> > (Please correct me if I'm wrong.) However, I still did not find any training related experiments extended to models other than CLIP and VIT (such as in Table 1 and Table 2). Due to this limited scope of experiment, and that the writing of this paper definitely needs to be improved, I'd like to remain my original score.

---

> > > ### Author Response · Authors · 2025-04-08
> > >
> > > Thank you for your follow-up and for acknowledging parts of the clarification.
> > >
> > > Regarding your remaining concern about training-related experiments beyond CLIP and ViT:
> > > **our main claim centers on representation dynamics**, specifically, the emergence of semantic abstraction as captured by ID. Fine-tuning in our study serves only to **support the interpretability-based explanation grounded in ID**, rather than being a primary focus of investigation. Accordingly, our additional experiments (**Figures A1–A5** in the appendix) focused on ID trends across a range of models, including ViT/L, ViT/H, MetaCLIP, SigLIP, and DINOv2, where consistent dynamics were observed.
> > >
> > > **Given your interest in training-related validation,** we have also conducted fine-tuning experiments on **MetaCLIP**, following the same setup as in Table 1 and Table 2. These results are now available in **Table A2 and A3** at https://anonymous.4open.science/r/TOT-078C/, and demonstrate **a similar robustness pattern**, further confirming the generality of our findings.
> > >
> > > We would also like to clarify that the experiments in Table 1 **do not involve fine-tuning**. They are inference-time evaluations designed to probe robustness under typographic perturbations. We mention this as the nature of the experiment appears to have been somewhat misinterpreted.
> > >
> > > We hope this additional information offers clarity. While we understand different reviewers may weigh emphasis differently, we believe the key contribution—**tracing the delayed and fragile emergence of semantics using ID**—is now supported by both theoretical insight and extended empirical results across architectures.
> > >
> > > We appreciate your time and feedback.

---

### Official Review · Reviewer_d3BZ · 2025-03-14

**Overall Recommendation:** 3

**Summary:**

This paper explores how vision-language models (e.g., CLIP) process text in images via the ToT (Textural or Textual) dataset, showing that early layers rely on visual texture while semantic understanding emerges in the final blocks. Using Intrinsic Dimension (ID) analysis, the paper reveals changing representational complexity across layers and propose fine-tuning the last layer to counter typographic attacks, yielding substantial performance gains in various defense scenarios.

**Claims And Evidence:**

The paper’s claim is clear and well-supported by experiments. Through the ToT dataset and ID analysis, the paper show how the model processes text and visuals across layers, culminating in stronger semantic understanding in the final layer.

**Weakness:**
1. The experiments were conducted only on ViT/B-16 and have not been verified on larger-scale ViT models (e.g., ViT/L-14, ViT/H-16) or other architecture (ResNet) of image encoders.

2. The paper classifies stages solely by changes in ID and accuracy. Could metrics commonly used in information bottleneck theory—such as mutual information—be adopted to further support the four-phase explanation of VLM image processing?

**Essential References Not Discussed:**

N/A

**Experimental Designs Or Analyses:**

The experimental design is solid and the analysis methods are effective. Through ID analysis and t-SNE, the authors show how representations evolve across layers and validate the defense strategy with fine-tuning.

**Weaknesses:**

1. The experimental could be expanded by including more model architectures to better validate the conclusions.
2. It is unclear whether all comparison baselines were fine-tuned on the ToT dataset, potentially affecting the reliability of the results. Specifically, it remains unclear whether performance improvements stem from higher-quality data or from the new fine-tuning strategy.

**Methods And Evaluation Criteria:**

The analysis and fine-tuning strategies fit the problem well. The ToT dataset carefully balances text semantics and visual forms, and metrics (accuracy, ID) effectively measure complexity and defense performance.

**Other Comments Or Suggestions:**

N/A

**Other Strengths And Weaknesses:**

**Strengths:**

1. The paper is well-written, with a clear motivation.
2. The design of the ToT dataset contributes a valuable resource for subsequent research in this area.

**Weaknesses:**

1. The conclusion that semantic understanding is delayed until the final layer lacks deeper theoretical explanation, and the available experiments only cover ViT-B/16. It remains uncertain whether the same pattern holds for larger ViT models.
2. There is no detailed training loss using in the fine-tuning.

**Questions For Authors:**

See above weaknesses.

**Relation To Broader Scientific Literature:**

This paper uses ID analysis to reveal changes in representational complexity across different layers, and proposes a defensce strategy against typographic attacks. However, the defence strategy is relatively simple to implement and shares core ideas with some existing work, suggesting limited novelty.

**Theoretical Claims:**

The paper lacks theoretical explanations, focusing on experimental results but not exploring why semantic understanding emerges late or how ID changes relate to it.

---

> ### Author Rebuttal · Authors · 2025-03-31
>
> We thank the reviewer for their close reading and thoughtful comments. We would like to clarify that this paper is primarily an **interpretability study** that analyzes how vision-language models process text in images, using Intrinsic Dimension (ID) as a lens to reveal the transition from textural perception to semantic abstraction across layers. Fine-tuning experiments are not our core contribution but serve to validate our representational insights. We intend our findings to serve not as a conclusion, but as a **foundation for deeper inquiries** into the representational dynamics of multimodal models.
>
> We address the reviewer’s concerns point-by-point below. Additional experimental results are available at: https://anonymous.4open.science/r/TOT-078C/
>
> ---
>
> **Q1. Generalizability beyond ViT-B/16**
>
> We appreciate the concern. Results on **ResNet** (Figure 9) are included in the appendix. Additionally, we evaluate **ViT-L/14, ViT-H/14, SigLIP, and MetaCLIP** in Figures A2–A5. Across all these architectures, we observe that significant shifts in ID values **consistently emerge in the final block**, suggesting that the phenomenon is not specific to ViT-B/16 but generalizes across a broad range of visual encoders.
>
> **Q2. Why not use mutual information or other theoretical metrics to support the stage transitions?**
>
> We understand the reviewer’s suggestion to explore more theoretical metrics such as mutual information for supporting the stage transitions. While MI is theoretically relevant, its estimation in high-dimensional, structured representations is **fundamentally limited by the bias-variance tradeoff of current estimators**. Methods like MINE or kNN often yield unstable or uninterpretable results across layers due to distributional shifts and the absence of ground-truth joint distributions. Given our focus on semantic emergence, ID offers a more stable and interpretable alternative.
>
> Our segmentation into representational "four stages" is not intended as a strict theoretical taxonomy, but rather as a **descriptive scaffold** that highlights where semantic abstraction emerges most distinctly. ID is used as a tractable and interpretable proxy for representational complexity, and it is especially well-suited to our goal of diagnosing typographic vulnerability. We view this as a methodological decision, **balancing interpretability with empirical rigor**.
>
> **Q3. Could the observed gains stem from data quality rather than the fine-tuning strategy?**
>
> To isolate the effect of fine-tuning, we introduced several controls:
>
> 1. In Appendix A.2, we replicate our ID analysis on a **distinct dataset** with matched structure. Results remain consistent, reinforcing that the observed gains are not tied to specific image content.
> 2. In Table 3, we conduct **cross-evaluation**, where each model is tested on all datasets (not just the one it was trained on).
> 3. All baselines in Table 4 were **fine-tuned on our datasets** to ensure fair comparisons under the same conditions.
>
> We believe these controls confirm that performance improvements arise from the **fine-tuning strategy**, not data artifacts.
>
> **Q4. The conclusion that semantic understanding is delayed until the final layer lacks deeper theoretical explanation**
>
> We believe the lack of a settled theory on when and how semantic understanding emerges in vision-language models reflects the field’s ongoing search for foundational insight, rather than a weakness.
>
> Our work contributes to this effort by introducing Intrinsic Dimension (ID) as a **structured, interpretable proxy** for representational complexity. We find that semantic fragility under typographic attacks coincides with sharp ID shifts in the final layers, suggesting a transition toward higher-order abstraction beyond surface features.
>
> While our interpretation remains exploratory, we argue that **such empirical signals are essential for building theoretical understanding**. By combining observations (Sec. 4), cross-model evidence (Sec. 5), and actionable metrics, we aim **not to close the question, but to make it visible and tractable**.
>
> **Q5. Clarify the fine-tuning loss**
>
> We use the **standard CLIP contrastive loss** for all fine-tuning experiments. The only modification lies in the construction of **positive pairs**, which we adapt to reflect semantic similarity across different attack settings. We do not introduce new loss functions. Figure B2 provides explicit training details.
>
> ---
>
> Thank you again for helping improve this work. We hope these clarifications address your concerns and highlight both the rigor and potential impact of our contributions.

---

> > ### Comment · Reviewer_d3BZ · 2025-04-04
> >
> > Thank you for providing the additional experiments and clarifications. Your response help the paper’s contributions and address my earlier concerns. Hence, I will raise my assessment score to 3.

---

> > > ### Author Response · Authors · 2025-04-04
> > >
> > > Thanks again for revisiting the score, we really appreciate your thoughtful engagement. We’re glad the responses were helpful and will clarify these points in the revision. Thanks for helping strengthen the paper.

---

### Decision · Program_Chairs · 2025-05-01

**Decision:**

Accept (poster)

**Comment:**

Even after discussion this paper has divergent reviews (2 weak accepts and 2 weak rejects). After reading the paper, reviews, and discussion, it is a very close decision. I tend to agree with the accepts that this task and analysis has inherent value and that the finding (in particular, that the last layer uniquely causes separation between texture and textual) is interesting. However, I also tend to agree with the rejects that the presentation of the paper could be improved and that there may be other possible explanations for the observations that would conflict with the conclusions being drawn.

While it's close, I advocate for acceptance. I think that the paper's task, dataset, and analysis make it useful to field and am sympathetic to the argument that in deep learning the best we can do is present multiple observations that are suggestive of a conclusion. I encourage the authors to integrate the reviewers suggestions on improving the readability and presentation of the paper.